# Dental Materials Applied to 3D and 4D Printing Technologies: A Review

**DOI:** 10.3390/polym15102405

**Published:** 2023-05-22

**Authors:** HongXin Cai, Xiaotong Xu, Xinyue Lu, Menghua Zhao, Qi Jia, Heng-Bo Jiang, Jae-Sung Kwon

**Affiliations:** 1Department and Research Institute of Dental Biomaterials and Bioengineering, Yonsei University College of Dentistry, Seoul 03722, Republic of Korea; caihongxin1@foxmail.com; 2The CONVERSATIONALIST Club, School of Stomatology, Shandong First Medical University, Jinan 250117, China; xiaotong011@outlook.com (X.X.); xinyuelu111@outlook.com (X.L.); z1924892050@outlook.com (M.Z.); aqms-happy@outlook.com (Q.J.)

**Keywords:** dental materials, CAD/CAM, 3D printing, additive manufacturing

## Abstract

As computer-aided design and computer-aided manufacturing (CAD/CAM) technologies have matured, three-dimensional (3D) printing materials suitable for dentistry have attracted considerable research interest, owing to their high efficiency and low cost for clinical treatment. Three-dimensional printing technology, also known as additive manufacturing, has developed rapidly over the last forty years, with gradual application in various fields from industry to dental sciences. Four-dimensional (4D) printing, defined as the fabrication of complex spontaneous structures that change over time in response to external stimuli in expected ways, includes the increasingly popular bioprinting. Existing 3D printing materials have varied characteristics and scopes of application; therefore, categorization is required. This review aims to classify, summarize, and discuss dental materials for 3D printing and 4D printing from a clinical perspective. Based on these, this review describes four major materials, i.e., polymers, metals, ceramics, and biomaterials. The manufacturing process of 3D printing and 4D printing materials, their characteristics, applicable printing technologies, and clinical application scope are described in detail. Furthermore, the development of composite materials for 3D printing is the main focus of future research, as combining multiple materials can improve the materials’ properties. Updates in material sciences play important roles in dentistry; hence, the emergence of newer materials are expected to promote further innovations in dentistry.

## 1. Introduction

The development and application of dental materials have a long history. Gold and silver, which were used by the Romans in 700 B.C. to repair cavities, are the first materials used for dental restoration. In the 18th century, wax was introduced to obtain impressions, which were then filled with a mixture of plaster and cementum to fabricate oral models [1]. Ceramics were then introduced into dentistry to manufacture porcelain restorations [2]. In the 20th century, composite resins, cement, titanium, stainless steel, and other materials were introduced into dental sciences [3]. Since the 21st century, computer-aided design and computer-aided manufacturing (CAD/CAM) has gradually gained popularity. Three-dimensional (3D) printing materials are more widely used today.

Three-dimensional printing technology, also known as additive manufacturing (AM), first appeared in the late 1970s. Three-dimensional printing gradually builds models by depositing materials layer by layer, which can be used for rapid prototyping and developing personalized 3D products using standardized materials based on CAD models. It uses CAD software to form a 3D digital model and transfer it to a 3D printer, which then converts the computerized digital model into layer-by-layer two-dimensional (2D) parts, generating solid layers to build the desired object [4,5,6,7,8,9,10]. At present, 3D printing has been widely used in many fields of life and has made significant progress [11]. Additive manufacturing technologies for 3D printing can be divided into 7 categories, based on how materials or inks are deposited: stereolithography (SLA), fused deposition modeling (FDM), digital light projection (DLP), selective laser sintering (SLS), computed axial lithography, photopolymer jetting, and powder binder printing [4,12].

In 2013, four-dimensional (4D) printing was proposed for the first time. Shape changing with time is the fourth dimension, besides the three dimensions of length, width, and height [13,14]. In addition, the shape, function, or properties of 4D printed objects can change under various stimuli, such as heat, light, electricity, and magnetic field [15]. Four-dimensional printing has advantages in terms of material adaptability, facilitating the precise configuration of material responsiveness. Further, the bioprinting included in it is an emerging technology, and its biggest advantage is the ability to create 3D structures of organisms (such as tissues, organs, nutrients, and cells), and the manufactured biological structures can change their functions [12]. Currently, 4D printing plays a major role in the medical field, mainly for tissue generation and transplantation. Four-dimensional printing, defined as the fabrication of complex spontaneous structures that change over time in response to external stimuli in expected ways, includes the increasingly popular 4D bioprinting [12,16]. Four-dimensional printing technology is roughly the same as 3D printing. Furthermore, 4D printing technology also involves inkjet bioprinting, extrusion-based bioprinting (EBB), SLA-based bioprinting, and laser-based bioprinting (LAB). Among them, inkjet printing and DIW are preferred, which are in their infancy and facing enormous challenges [17,18,19,20,21,22]. 

Commonly used materials for dental 3D printing include polymers, ceramics, and metals, most of which possess high accuracies, excellent biocompatibilities, and good mechanical properties. Materials used for 4D printing mainly include Shape Memory Polymers (SMPs) and hydrogels. SMPs include thermally induced SMP, photochromic SMP, and chemical induction SMP [12]. Four-dimensional printing materials are summarized as synthetic polymers (polylactic acid (PLA), acrylonitrile butadiene styrene (ABS), polyvinyl alcohol (PVA), polycaprolactone (PCL), polymethyl methacrylate (PMMA), etc.) and natural biopolymers (hyaluronic acid (HA), chitosan, alginate, etc.), and related cells and growth factors work with them. 

However, depending on their specific principles, 3D/4D printing technologies can individually adapt to different materials. Given the rapid advances in digital technologies and specific clinical requirements, an overview of printing materials could be helpful. This review aims to list contemporary 3D printing and 4D printing materials and describe their properties, current stage of development, features, applicable 3D/4D printing technologies, and corresponding indications and applications in dentistry. Emerging 3D-printing dental materials with noteworthy potential for future developments are also discussed. Table 1 lists the differences between 3D printing and 4D printing in terms of technology, materials, applications, and so on.

## 2. Three-dimensional Printing Materials

### 2.1. Polymers

Polymers are the most popular materials used for dental and maxillofacial surgical 3D printing. They can be used to manufacture surgical guides, custom trays, working casts, and temporary restorations, supported by implants, crowns, and bridges [23,24]. Polycaprolactone (PCL), polymethyl methacrylate (PMMA), polylactic acid (PLA), poly (lactic-co-glycolic acid) (PLGA), and ultraviolet (UV) resins are some of the representative polymers used for 3D printing in dentistry. 

#### 2.1.1. PCL

As an aliphatic polyester, PCL has superior biocompatibility, adjustable degradability, and wide applicability in the biomedical field. A lower melting point (59–64 °C) and excellent compatibility with other substances that are easy to manufacture make it a suitable substitute for materials processed by FDM [25]. PCL is stable in vivo, owing to its high hydrophilicity and solvent solubility. During the process of melt extrusion in 3D printing, its crystallinity and crystal directionality are developed [26].

PCL and its composites are widely used in tissue engineering [27]. The mechanical properties of such scaffolds are affected by many factors, such as composition and porosity [28]. PCL functions as a membrane for guided bone regeneration, and its biocompatibility can be improved by modification. Compared with other printing materials, PCL, with a lower melting point, is considered to be more biocompatible, for it is less likely to cause inflammatory reactions [29,30,31]. Bone defects implanted with membrane-loaded human recombinant bone morphogenetic protein-2 (rhBMP-2) have been shown to achieve calvaria defect healing in 8 weeks [32]. The porosities of the membrane and scaffolds are important influencing factors of bone-healing [33,34]. The designed pores provide space for cell proliferation. A mixture containing bioactive materials, such as hydroxyapatite, β-tricalcium phosphate, and rhBMP-2, enhances biocompatibility and promotes new bone formation in in vivo experiments [35,36]. Three-dimensionally printed PCL scaffolds have been used in bone tissue regeneration (Figure 1A) [37].

#### 2.1.2. PMMA

PMMA, which was first discovered in the 1930s [41], is a commonly used polymer in dentistry. Given its advantages, such as ease of manufacturing, lower cost, and stability in the oral environment, PMMA is the most popular base material for dentures. It also functions as bone cement for screw fixation in the bone, can fill bone defects in cavities and the skull, and can be used to stabilize vertebrae [42]. Other applications include temporary crowns and bridges, obturators, and retainers in orthodontics [43]. Even though PMMA has been applied to prosthetic restorations with predictable clinical performance, the interactions between the oral environment and PMMA are not fully studied by scientists, and potential side impacts on cells have been discovered [44]. Compared with polyamide-12, PMMA can maintain its color as a denture base material [45]. Nanodiamonds in low concentrations can be added as reinforcement to improve the overall properties of PMMA-based fixed interim prostheses [46]. PMMA can be manufactured by FDM [47]. One study also found that SLA technology could be used to print PMMA composites with three different reinforcements: aluminum nitride, titanium oxide, and barium titanate [48]. However, the limited number of studies on 3D printed PMMA compositions and biocompatibility require further investigations. Even though the mechanical properties of 3D printed PMMA are worse than those of PMMA machined by traditional methods, the optimized parameters, as well as infiltration of epoxy and smaller particles, may influence the qualities of the final products [49]. As shown in Figure 1B, 3D printed PMMA functions as a provisional crown after accurate design [38].

#### 2.1.3. PLA

Polylactic acid (PLA) is also a hydrophilic and aliphatic polyester, which is biodegradable and environmentally friendly, making it an ideal material for 3D printing in the future. PLA can also be manufactured by FDM. The process of 3D printing leads to lower molecular weight and degradation temperature, without changes in the semicrystalline polymer structures [50]. 

However, deformation will occur when the environmental temperature exceeds 50 °C, which may limit its application in 3D printing. Therefore, the idea of forming hybrid materials with other polymers arises. When combined with other bioactive substances, PLA-integrated guided tissue and guided bone regeneration (GTR/GBR) membrane reinforced by magnesium has the best charge capacity, corrosion resistance, and cell adsorption [51]. It has been found that PLA scaffolds combined with nanohydroxyapatite can function as carriers for cells in clinical dental treatment [52]. Deng et al. found that maxillary complete dentures made by FDM with PLA fulfill the need for accuracy [53]. Three-dimensionally printed PLA biodegradable scaffolds were used to produce cell-derived decellularized matrices; this has increased the applicability of 3D printing technology in dental regenerative medicine [54]. Three-dimensionally printed PLA functions as a drilling guide for oral implantology [55]. In general, PLA is an ideal material, with good clinical prospects. PLA can also be considered as a raw material for 3D printed bio-scaffolds for hard tissue, with apatite consisting of calcium and phosphate formed on its surface, after being immersed in simulated body fluid solution for a certain period (Figure 1C) [39].

#### 2.1.4. PLGA

PLGA, which is a copolymer of PLA and polyglycolic acid (PGA), is considered an ideal material, owing to its biodegradability and biocompatibility. Fabricated using a variety of methods, PLGA-based composites combined with metal reinforcements have vast dental application prospects. The composites allow diverse properties, including enhanced cell proliferation and improved antimicrobial properties [56]. The physicochemical properties of PLGA allow manufacturing by FDM. The optimized 3D printing parameters vary for copolymers formed with various ratios of PLA and PGA, which influence the molecular weight, end cap of the copolymer, and physiochemical properties, such as viscosity and heat flow under various temperatures, and a higher proportion of PLA in PLGA is recommended for 3D printing [57]. 

Three-dimensionally printed PLGA scaffolds have been introduced in tissue engineering. Mironov et al. investigated novel bioresorbable scaffolds by 3D printing based on the PLGA, and it was observed that these scaffolds were non-cytotoxic, demonstrated cell proliferation in the chosen stem cells, and had excellent viscosity [58]. When repairing bone defects, a novel porous scaffold containing PLGA, tricalcium phosphate (TCP), and Mg powder was manufactured using low-temperature rapid prototyping technology, and it has been demonstrated that this newly designed scaffold has obvious biosecurity, promotes new bone and vessel formation, and improves bone tissue quality in rabbit models [59]. The in vivo characteristics of 3D printed PLGA scaffolds have also been assessed. Excellent osteoconductivity and biocompatibility have been observed in the periosteum and in animals with iliac defects [60]. Biological membranes can thus be manufactured by 3D printed PLGA (Figure 1D) [40].

#### 2.1.5. UV Resin

UV resins consist of a polymer monomer, a prepolymer, an active diluent, a photoinitiator, and a photosensitizer [61]. Under UV light of 250–300 nm wavelength, polymerization reaction commences immediately, and the scattered polymers combine into crosslinked polymers, turning from a liquid polymer resin to a solid 3D structure [62]. The conversion degree is affected by the monomer type, photoinitiator type, temperature, and light intensity [63]. UV resins are basic and essential materials for DLP and SLA [64]. The applications of UV resin include crowns, bridges, surgical guides, and other prostheses [65,66]. 

UV resins have high curing efficiencies, low energy consumption, low solvent emissions, low cost, and moderate curing conditions. Titanium dioxide and tartrazine lake affect the viscosity and curing thickness of the resin [67]. However, sunlight may change the shape and color of the product printed with SLA, because of the polymers’ sensitivity to UV light.

Table 2 lists the above-mentioned polymers currently used for 3D printing and their corresponding characteristics, applicable technologies, and clinical applications.

### 2.2. Metals

With their ideal mechanical and biological properties, metals have always been a good choice for dental restoration materials, mainly titanium (Ti) and cobalt–chromium (Co–Cr) alloys. However, their esthetic properties and higher cost render them less than optimal for use [68,69]. 

#### 2.2.1. Ti and Its Alloys

Ti and its alloys are chemically active. Chemically stable oxide films, which are dense and strongly adherent, can be formed on their surfaces in air and other media to protect the substrates from corrosion. Their excellent mechanical properties, outstanding biocompatibility, and low density make them ideal materials for 3D printing [70]. Over the past 20 years, SLM, SLS, EBM, and DMLS have been introduced as promising manufacturing technologies for Ti and its alloys [71,72,73,74,75,76,77]. Owing to their excellent reproducibility, flexibility in design, relatively good resolution, and low-cost effectiveness, these 3D printing methods are widely used to manufacture implantable devices (i.e., dental implants and scaffolds) and prosthetic devices (i.e., dental crowns and denture frameworks) based on Ti and its alloys [69,78,79,80]. Studies have shown that Ti and Ti alloy samples printed by optimized 3D printing technologies have comparable biological properties to machine-milled products [72,81,82]. Yang et al. prepared 3D printed porous Ti6Al4V dental implants and applied a chitosan-based composite coating, which is non-toxic, is favorable for cell proliferation, and has good mechanical properties, thus contributing to the growth of new bone to support damaged bone [81]. Despite their unique advantages, Ti and its alloys can still corrode after implantation because of chewing forces, acidic environments, and so on [83]. Furthermore, 3D printed Ti alloys have shortcomings, such as high cost, poor wear resistance, easy embrittlement, and potential toxicity [4].

At present, medical devices based on Ti and its alloys have been used to fabricate personalized mandibular implants for maxillofacial surgery (Figure 2A) [84].

#### 2.2.2. Co–Cr Alloys

The good mechanical properties and corrosion resistance of Co–Cr alloy allow it to be used to create products with high strength and high-temperature resistance [86]. Co–Cr alloy is hence considered an ideal material for fabricating prostheses and non-precious metal frameworks [82,87,88,89,90,91,92,93]. 

Co–Cr alloys can be printed by SLA, SLM, and DMLS, allowing rapid and accurate melting of the metal powders into layers using high-powered laser beams, and, consequently, the direct printing of products [69,73,86,91,94,95,96,97,98]. Numerous studies have shown that 3D printed Co–Cr alloy restoration not only meets the needs of dental practices (e.g., acceptable marginal clearance, higher biocompatibility, excellent microstructural homogeneity, and metal-ceramic bond strength) [69,95,99,100], but also eliminates the challenges associated with the milling process of Co–Cr alloys, such as shrinkage of the materials during the casting process. Owing to their improved properties, Co–Cr alloys are promising alternatives for the construction of dental restorations [24,96,101,102,103,104]. Furthermore, the post-production heat treatment of Co–Cr alloys made by SLM allows the effective release of the residual stresses, resulting in a homogeneous microstructure and improved mechanical properties [105]. Unfortunately, the printing parameters, melting temperature, powder adhesion, slagging, thermal stress accumulation, residual stress, and other constraints inherent to SLM technology limit the performance and hence the quality of the 3D printed products [69,95,106]. In prosthodontics, DMLS technology can be applied to manufacture Co–Cr alloy prostheses (Figure 2B) [85].

#### 2.2.3. Others

Stainless-steel and magnesium alloys, both of which have good physical properties and biocompatibility, can be used as laser-selective melt-forming metals. Stainless steel has good sterilization capabilities and can be used for rapid prototyping of the dental implants and orthodontic components by SLM [107,108]. Magnesium (Mg) and its alloys may have important functional roles in the physiological system, due to their close mechanical properties to human bone tissue, their natural ion content, and their in vivo biodegradation characteristics in body fluids. They are usually used as an implant for orthopedic and dental treatment [109]. They also have good mechanical properties and biocompatibility and are promising for oral implantology. Julia et al. successfully modified magnesium implants through a high-pressure anodic oxidation process to obtain biodegradable magnesium-based implants. They are a new type of biomaterial with potential for dental applications and have the potential to make dental metal medical devices [110]. Zhang et al. evaluated magnesium alloys created using SLM technology to implant the fabricating materials for manufacturing porous structural implants, and the mechanical properties of the composite porous implants met the requirements of living for edentulous patients [111].

In addition, new materials, such as nanomaterials, which are less than 100 nm in diameter, have been introduced in dentistry. Adding nanometals and their oxides to other nanomaterials can enhance the antimicrobial, mechanical, and regenerative properties. Metal elements, such as Ag, Cu, Au, Ti, and Zn are antibacterial. Nanometals, such as Ag, Au, ZrO2, and TiO improve orthodontic adhesive’s compressive, tensile, and shear bond strengths. The question of whether nanometals have adverse effects on the oral system still needs further research [112,113,114].

The above-mentioned commonly used metals and their relevant information are listed in Table 3.

### 2.3. Ceramics

Ceramic materials have become ideal products for dental restorations because of their excellent mechanical properties, biocompatibility, good abrasion and corrosion resistance, and esthetic characteristics, similar to natural teeth [115,116,117]. The commonly used 3D printed ceramic materials in dentistry can be divided into glass, zirconia, and alumina ceramics, based on composition [118,119]. 

#### 2.3.1. Glass Ceramics

Glass ceramics are a class of composite materials made via high-temperature sintering, molding, and heat treatment, combining crystalline phases. Dental glass ceramics were first proposed as a mica-based material by Malament and Grossman in the mid-1980s [120]. In the 1990s, scientists developed a stronger and more reliable microcrystalline glass for better dental restorations [121,122]. 

Glass ceramics have high mechanical strengths, low electrical conductivities, high dielectric constants, high chemical resistances, high thermal stabilities, and other superior properties [123]. Depending on their application, dental glass ceramics can be divided into bioactive dental glass ceramics (BDGCs), which have dental bonding capabilities and stimulate specific biological responses at the material/tissue interfaces, and restorative dental glass ceramics (RDGCs). BDGCs are suitable for hypersensitivity therapy, implant coatings, bone regeneration, and periodontal healing [76], while RDGCs can be applied to manufacturing inlays (Figure 3A), onlays, full crowns, partial crowns, bridges, and veneers [124]. Recently, lithography-based AM [125] and SLA [126] have been widely used to fabricate glass–ceramic restorations. Experiments have shown that glass ceramics with excellent mechanical properties can be effectively obtained using the SLA 3D printing process with appropriate parameters [127,128]. The exposure intensity in the SLA process affects the curing width and curing depth, as well as the quality of the final product [127].

#### 2.3.2. Zirconia

Zirconia occupies an important position among oxide ceramics and is the current focus in dental materials, with a wide range of applications [131]. From a materials-science perspective, zirconia crystals are divided into three structures: monoclinic phase (m), tetragonal phase (t), and cubic phase (c), among which m is the most common form of zirconia at room temperature. However, when the temperature changes, these three structures can be interconverted [132,133]. Zirconia ceramics commonly used in dentistry are 3 mol% yttria-stabilized tetragonal zirconia polycrystal (3Y-TZP) [134], zirconia-toughened ceramics, partially stabilized zirconia ceramics, and nano-zirconia and alumina-composite ceramics. Each of these four types of zirconia ceramics contains a stable tetragonal phase and is toughened by the martensitic phase transformation of the t-phase with varying microstructure, resulting in different properties and processing techniques. 

Zirconia is biocompatible and osteoconductive, and accommodates the surrounding soft tissues to facilitate bone formation [135]. Specifically, zirconium can reduce inflammatory responses, plaque accumulation, and bacterial population, thereby altering fibroblast adhesion and proliferation [136]. In terms of mechanical properties, zirconia is known as “ceramic steel”, as it has good toughness, ideal hardness, and high strength [137,138]. Zirconia plays an important role in prosthetic dentistry and is the raw material for root-canal piles, crown and bridge restorations, and implant abutments [139]. Zirconia, especially Y-TZP, has high toughness and strength, and has become the most durable among dental ceramics, but its esthetics are suboptimal, and the current Y-TZP zirconia suitable for anterior prostheses is constantly developing and improving [140,141]. 

SLA/SLS was introduced to fabricate zirconia, reducing the time of the restorative system fabrication and improving the treatment efficiency [142]. High internal stresses, cracks after sintering, and high-volume shrinkage may affect the mechanical properties and clinical suitability of 3D printed materials. Zirconia crown restoration manufactured by 3D printing is shown in Figure 3B [129].

#### 2.3.3. Alumina

Aluminum oxide, or alumina (Al2O3), was first introduced in the 1970s as a ceramic material obtained by calcining aluminum hydroxide, which is often referred to as alumina trihydrate (ATH) [4,143,144]. Alumina includes monocrystalline and polycrystalline alumina in several forms: α, χ, η, δ, κ, θ, γ, and ρ. Alumina has high mechanical strength and excellent chemical stability. The higher purity usually results in higher strength [145]. To reduce the possibility of fracture, zirconia particles are often added to the matrix to obtain zirconia-toughened alumina (ZTA) ceramics with optimal strengths, high hardness, and good stabilities [146]. Cut or milled ceramics are preferred for obtaining flat surfaces. With the development of AM, 3D printed ceramics are gradually meeting complex shape requirements [147,148]. Alumina ceramics can be produced using inkjet printing technology. It is worth noting that the powder’s dimensions are critical, owing to the small size of the nozzle. Recently, it has been found that alumina ceramics can be used in FDM technology, while printing the smallest overhangs or cavities is not possible with milling technology [143,147,149]. One study reported that the physical and mechanical properties of alumina ceramic samples manufactured using SLA technology are comparable to those fabricated using conventional methods [150]. Another study used SLA technology to create precise alumina crown frameworks and have found that it can reduce the marginal gap between the crown framework and the preparation of the crown by refining the SLA manufacturing process. Figure 3C shows an alumina crown frame printed by SLA technology, placed on an all-ceramic crown preparation [130].

Some basic information on above-mentioned ceramics is presented in Table 4.

## 3. Four-dimensional Printing Materials

Four-dimensional printing materials refer to cell preparations containing biomaterials and bioactive ingredients, which create conditions for the applications of 4D printing in regenerative dentistry [151]. The process of 4D printing can be divided into two categories, according to whether scaffolds are needed, including scaffolds based on cells and the scaffold-free cell-based approach, directly printing cells to form new tissues [152,153,154]. Four-dimensional printing materials are composed of biological materials and living cells. After a certain period, the scaffold degrades and the cells grow in a predetermined way, which shows the requirement that bioinks have certain biocompatibility and self-preservation capacity. They can also be used in tissue regeneration engineering, such as bone and tooth tissue regeneration, with rapid gelation kinetics, tunable mechanical properties, good biocompatibility, and tissue adhesion [155]. Four-dimensional printing materials are summarized as synthetic polymers and natural biopolymers, and related cells and growth factors work with them.

### 3.1. Polymers

#### 3.1.1. Synthetic Polymers

Synthetic polymers have been widely used in biological 3D printing [156], including PCL, PLA, and PLGA, as mentioned in the previous section [151,157]. In recent years, as the most widely used aliphatic polymer, PCL has gained an important position in the medical field and has potentially become a candidate material for craniofacial repair. Pluronic and poly (ethylene glycol) (PEG) is also a promising material in 4D printing. Pluronic is a block copolymer, consisting of two hydrophobic groups with a hydrophilic group between. Pluronic is capable of forming self-assembled gels at room temperature and is mobile at 10 °C [158]. Forming bioinks with other different polymers, PEG increases the mechanical properties of the origin structures [152,159].

Synthetic polymers are low-cost, mass-producible, chemically stable, and have appropriate degradation rates and photo-crosslinking capabilities [151,160,161]. Although synthetic polymers are not as biocompatible as natural polymers, they are tunable, have a lower gel temperature, and can withstand temperature and pH changes [151,152,162]. Even though, compared with natural polymers and bioceramics, aliphatic ethers have much lower absorption rates and less promotion on cell adhesion [163]. Adding synthetic polymers to natural ones gives more stable structures with tunable 4D printing properties, such as suitable porosity, surface area, and mechanical strength [152,163]. Moreover, synthetic polymers can now mimic the amphiphilic characteristics of natural polymers and introduce antimicrobial properties [164].

#### 3.1.2. Natural Biopolymers 

Biopolymer-based composite materials are biocompatible and biodegradable [165]. Natural polymers are usually important materials for fabricating bioinks, the advantages of which are that they can mimic the structure, self-assembly, and biocompatibility of the natural extracellular matrix (ECM). Common natural polymers, including hyaluronic acid (HA), collagen, agarose, chitosan, alginate, etc. have been introduce to 4D printing. 

HA, a natural linear polymer, belongs to the ECM, and can usually be isolated from human or animal cartilage or connective tissue [166,167]. Presently, there are many HA-based bioinks, such as HA-based hydrogel bioinks. A recent study has shown that the printed products have high mechanical properties and stability [168]. Hydrogel has excellent biocompatibility, mechanical strength, biological and chemical properties, which made it the most common biological ink material [169]. Hydrogel simulates the microenvironment of the extracellular matrix, and is conducive to cell attachment, proliferation, and differentiation [170]. The modification of hydrogel by methacrylate improves the osteogenesis [168]. Photo-crosslinking HA bioink, chemical-crosslinking HA bioink, and HA-based double-crosslinking bioink have also been applied to 4D printing, and their rheological properties, mechanical properties, and the amount of cell adherence are improved after chemical modification and the addition of cell adhesion oligopeptides [171,172]. Associated with other polymers, such as gelatin and carboxymethylcellulose gel and thiol, HA-based bioinks have enhanced cell viability and biological stability [173,174]. HA promotes tissue repair and wound healing, facilitates recovery from dental surgery, and is also effective in periodontitis and gingivitis therapy [151,175].

Collagen, which also belongs to the ECM, can be used to make bioink, both separately or in combination [176]. Because its crosslinking or gel requires specific temperature and time, the effect of mixing collagen and other materials for printing is conducive to printing collagen separately [152,177]. For example, the combination of collagen and alginate is conducive to cell attachment, improves the mechanical properties of printing products, and is conducive to the application in cartilage tissue engineering [178]. Collagen, combined with gelatin, can improve biological activity and rheological properties [179].

Agarose is a marine polysaccharide whose main chain is composed of disaccharides. Although it has good gel properties, mechanical properties, and biocompatibility, it has limited ability to support cell growth. In order to enhance its performance, biological materials, such as collagen, fibrinogen, and sodium alginate, are often added to agarose gel as agar-based bioink [178,180]. The results show that these materials can promote cell growth and enhance the mechanical properties of bone tissue. In order to maintain the stability of the printing structure and enhance the various properties of cells, chemical treatment, such as carboxylation, is required [178,181,182,183].

Chitosan, a natural biopolymer, plays an important role in dentistry and has been widely used in prosthodontics, oral implantology, and endodontics [184]. Chitosan is not easily soluble in water, but is soluble in acids with pH lower than 6.2. Although it has good biocompatibility and renewable ability, its low mechanical strength limits its application in hard tissue regeneration, and its stability and mechanical properties can be enhanced by adding calcium ion (Ca^2+^) [185].

In dentistry, alginate, which is a natural polysaccharide, is generally used to obtain conventional impressions. The alginate biopolymers are suitable for making bioinks, because they can adsorb other molecules and water, which diffuse outwards [186]. Alginate-based bioinks can print 3D structures, both with or without chondrocytes, at the same time. Microchannels printed with sodium alginate-based hydrogel materials have higher strength. Polymers such as PCL, gelatin, and poloxamer can also be mixed with sodium alginate for printing [187,188,189,190]. 

Cellulose, derived from plant fibers, can be used to make tissue engineering scaffolds and wound dressings, and plays a significant role in drug delivery [191]. It has hydrophilic, as well as good insulating and anti-electrostatic, properties, making it the most abundant, renewable, and stable natural linear polymer, widely used in industrial and biomedical fields [180,192,193]. Because of its biocompatibility, different types of cellulosic materials, such as nanocrystals (CNCs), cellulose nanofibers, and nanofibers (CNFs), are often added to the hydrogels to improve the viscosity of bioinks and ensure higher printing accuracy [173,194,195]. The combination of cellulose with alginate, hyaluronic acid, PLA, and other polymers to produce printable hydrogels is the key to the use of cellulose in tissue engineering to print 3D scaffolds and other structures [195]. Adding lignocellulosic fibers to other biopolymers can significantly improve the mechanical strength of biopolymers [180].

The basic information about the above-mentioned natural biopolymers is listed in Table 5.

### 3.2. Cells 

Cells are essential parts in bioinks for guiding tissue and organ regeneration. Mesenchymal stem cells (MSCs) with immune adjustability properties and osteoblasts are ideal cells for 4D printing. Cells can be printed with or without base biomaterials that provide scaffolds. Cell aggregates and spheroids can be directly introduced as 4D printing materials. Cell aggregates refer to the bioinks based merely on cells, without any additional biomaterials.

Multiple cells have been applied to 4D printing in dental and maxilla-facial surgery (Table 6 [201,202,203,204,205,206,207,208,209,210,211,212,213,214,215,216,217,218,219,220,221,222]), including both dental and non-dental cells. Dental pulp stem cells (DPSCs), stem cells from the apical papillae (SCAPs), periodontal ligament stem cells (PDLSCs), dental follicle cells (DFCs), gingival fibroblasts (GFs), and stem cells from human exfoliated deciduous teeth (SHEDs) have been applied to 4D printing in tooth regeneration. Non-dental cells include bone marrow stem cells, pre-osteoblasts MC3T3-E1 cells, and MSCs from gingival tissue. The adjustment of differentiation of DPSCs can be achieved by bioactive ingredients, such as concentrated growth factors [223,224]. Similar to DPSCs, SCAPs show a strong mineralization capacity, with a high expression of dentin sialophosphoprotein, which reflects the odontogenic differentiation [225]. The 4D printing of periodontal ligament cells can improve biocompatibility, which can be applied to periodontal therapy, especially pulp revascularization [226,227]. DFCs, which surround the tooth germ, are derived from mesenchymal connective tissue, and have the best proliferation capacity among all the mentioned cells and possible differentiation to osteoblasts and cementoblasts [211]. GFs, with different phenotypes, have therapeutic potential for regenerative medicine. The functionalized base biomaterials trigger the differentiation and proliferation of the GFs. In a clinical study, it has been found that culture GFs significantly reduce vertical pocket depth, compared with β-calcium triphosphate and collagen membrane in the treatment of intrabony periodontal defects [228]. The biocompatibility between hybrid polymers membrane and GFs has shown that GFs are promising cell materials for bioinks [229]. Being extracted from the lost deciduous tooth, SHEDs can promote bone formation and produce dentin in in vivo experiments and are able to improve the expression of HA compared with human fibroblasts, which demonstrated that SHEDs could provide a new strategy for wound healing [230,231].

Four-dimensional printing technology applied in dental and cranial facial surgery consists of the bone regeneration, periodontal complex, dentin-pulp complex (Figure 4), pulps, neural regeneration, whole tooth regeneration, and cartilage, which is mainly applied in the temporomandibular joint [232,233]. However, the segmentation during 3D printing would be a problem to be solved, resulting in the instability of the printing process, leading to the clog of the nozzle and the printability of the products [234]. Currently, researchers have studied the immunological properties of different biomarkers on the surface of stem cells’ membranes. The correlation between biomaterials and stem cells is, today, a research hotspot, and the surrounding factors, such as local pH and metal ions’ impacts on the stem cells, are also required for deeper investigation.

### 3.3. Growth Factors

Growth factor is a soluble signaling molecule that binds stably to ECM and controls a variety of cellular responses, such as cell growth, value addition, and differentiation. The selective release of growth factors is essential for the bioactivity and tissue regenerative capacity of bioinks. Suitable materials, such as natural and synthetic polymers, are usually selected to incorporate growth factors into sponges, micro/nanoparticles, and hydrogels [235,236]. Recently, the incorporation of cells and growth factors into bioinks to promote bone tissue regeneration has been achieved.

Stromal-derived factor 1 (SDF-1) has a role in regulating cell migration and cell growth, and promotes vascular regeneration and bone regeneration [237]. It can be incorporated into hydrogels for preservation. The addition of SDF-1 to PLA promotes the migration of bone regeneration cells into the printed scaffold and promotes osteogenesis [238]. Bone morphogenetic proteins (BMP) have the ability to induce bone regeneration and bone healing. Bone scaffolds printed with bioinks, and supplemented with BMP-2 and transforming growth factor-beta1 (TGF-β1), stimulated bioactivity and significantly improved bone regeneration [239]. The co-assembly of hydrogels, SDF-1 and BMP-2, achieved a controlled release of both growth factors and promoted the regeneration of periodontal bone tissue [240]. BMP-7 was added to PCL and β-TCP to print bone scaffolds to repair bone defects in animal models, and the results showed better good tissue ingrowth and bone regeneration [241]. BMP-7 was added to PLA to print bone scaffolds for repairing defects and analyzed for bone regeneration. The results showed that BMP-7 was more effective in bone regeneration treatment and the newly generated bone had a porous structure [238]. Vascular endothelial growth factor (VEGF) regulates cell migration, angiogenesis, and bone regeneration [239]. Growth factors (e.g., VEGF and BMP-2) can also bind to nanoparticles to control the rate of growth factor release through nanoparticle movement and integration [242].

Above-mentioned growth factors are listed in Table 7.

## 4. Futural Prospects and Conclusions Marks

In summary, with digital technology’s development, 3D printing and 4D printing have been adopted in oral clinical medicine. Due to the uniqueness in individual personalization, 3D printing has gained wide application in dentistry. Traditionally applied materials include polymers, metals, and ceramics. Three-dimensional bioprinting, also known as 4D printing, has been developed to improve the therapeutic effects in dento-maxillo-facial surgery. Four-dimensional printing introduced both synthetic and bio-natural materials as raw materials and stem cells, which accelerates the healing period after surgery and achieves bioactive repairing after surgeries.

Futural studies will continue to focus on the regulation of both biological and physical properties of printed products, with less possibility of inflammation after surgeries and induction of early bio-reaction. Presently, to improve the properties of dental materials, researchers are now studying the correlation between each material and working on the investigation of composite materials with desired properties and characteristics. Hydroxyapatite has been corporate into polymers to improve osteoinductivity. Different surface modifications have been studied on the metal to improve the antibacterial properties of titanium and its alloy, and protective coatings were manufactured on the surface of magnesium alloy to adjust the degradation rate. Also, to optimize the bioactivities of the 4D printed products, the biological actions, including stem cell proliferation and differentiation regulation, and the interaction of biomaterials with stem cells have been extensively studied.

It is believed that, with the rapid development of material investigations, multiple ideal novel materials and techniques, with better biocompatibility and bioactivity, would be applied clinically to contribute to the prosperity of dental practices.

## Figures and Tables

**Figure 1 polymers-15-02405-f001:**
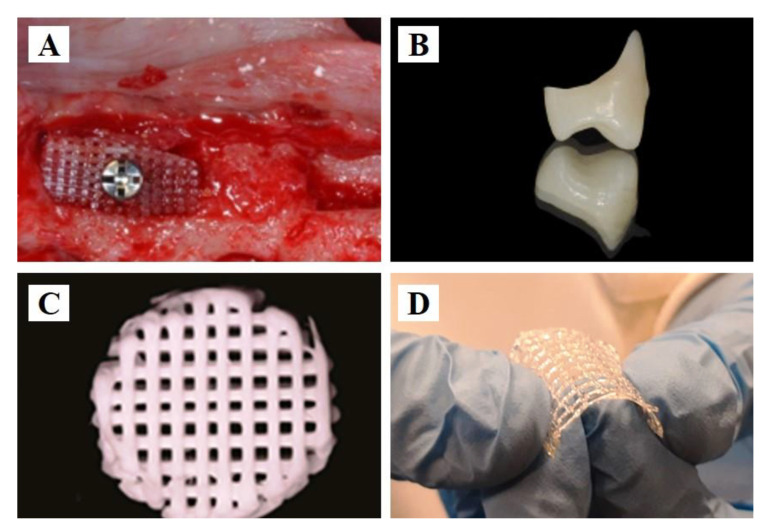
Polymer applications in 3D printing dentistry. (**A**) Bioengineering scaffold manufactured from PCL in a dog’s mandible. Reprinted with permission from Ref. [37]. (**B**) Printed PMMA for the provisional crown. Reprinted with permission from Ref. [38]. Copyright 2022 Elsevier. (**C**) PLA 3D printed scaffold. Reprinted with permission from Ref. [39]. (**D**) PLGA low-temperature solvent-based 3D printed material for biological membrane. Reprinted with permission from Ref. [40]. Copyright 2020 Taylor & Francis.

**Figure 2 polymers-15-02405-f002:**
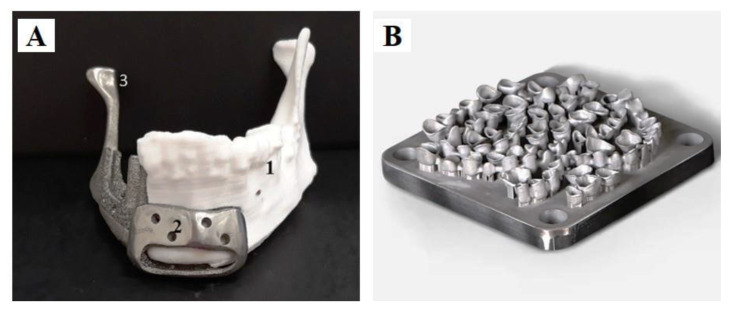
3D-printed metals in dentistry. (**A**) EBM-manufactured Ti mandibular implant. Reprinted with permission from Ref. [84]. Copyright 2018 Springer Nature. (**B**) Co–Cr prosthesis manufactured by DMLS. Reprinted with permission from Ref. [85].

**Figure 3 polymers-15-02405-f003:**
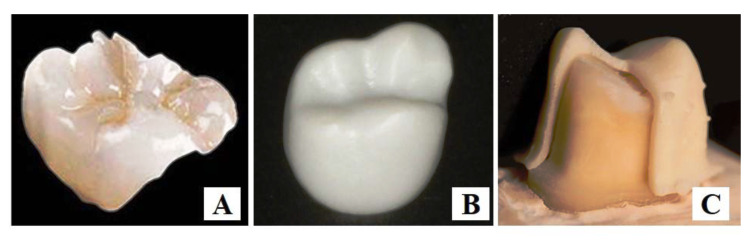
Ceramics for prosthetic dentistry. (**A**) Bioactive glass inlay. Reprinted with permission from Ref. [124]. Copyright 2016 Wiley. (**B**) Zirconia crown restoration [129]. (**C**) SLA-printed alumina crown. Reprinted with permission from Ref. [130]. Copyright 2017 Elsevier.

**Figure 4 polymers-15-02405-f004:**
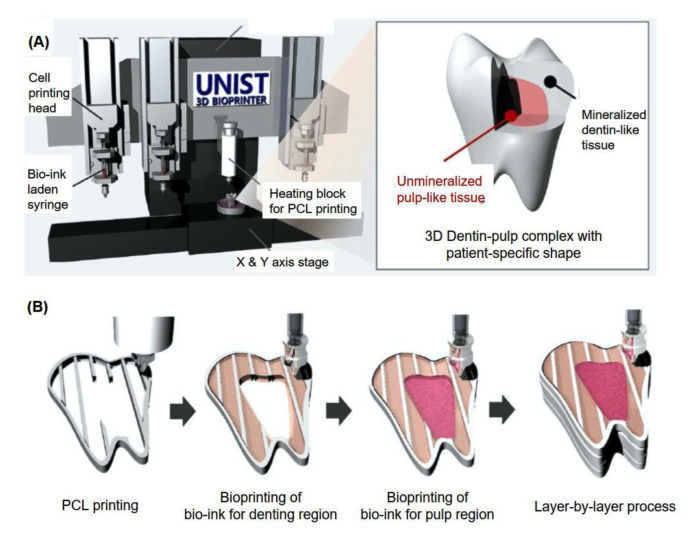
Bioprinting process of patient-specific shaped 3D dentin–pulp complexes. The illustrations show schematic drawings of the (**A**) 3D bioprinter and (**B**) printing process to produce patient-specific shaped 3D dentin–pulp complexes. The complex was constructed by serial printing of PCL and two bioinks for dentin and pulp tissue in a layer-by-layer manner. Reprinted with permission from Ref. [232]. Copyright 2019 SAGE Publications.

**Table 1 polymers-15-02405-t001:** The differences between 3D printing and 4D printing.

	3D Printing	4D Printing
Technology	SLA, DLP, FDM, SLS, photopolymer jetting, powder binder printer, and computed axial lithography.	FDM, SLA, DLP, direct ink writing, inkjet [12].
Suitable materials	Thermoplastics, metals, ceramics, biomaterials, or nanomaterials [13].	Self-assembled materials, multi-materials, designed materials [13].
Applications	Jewelry, toys, fashion, entertainment, automobile, aerospace, defense, biomedical devices, etc.	Soft robots, grippers, drug delivery, stent and tissue engineering, etc. [12]
Advantages	High material utilization and the ability to manufacture a single complex geometry [6].	The precise configuration of material responsiveness [12].
Disadvantages	Time-consuming post-processing, static microstructure, limited layer-by-layer printing speed [12].	Slow response rate and low efficiency [12].

**Table 2 polymers-15-02405-t002:** Polymers for 3D printing in dentistry.

Abbreviation	Full Name of Polymer	Applicable 3D Printing Technologies	Characteristics	Clinical Applications
PCL	Polycaprolactone	FDM	Superior biocompatibility and adjustable degradability	Tissue engineering scaffolds [27,29,30]
PMMA	Polymethyl methacrylate	FDM/SLA	Easy to manufacture, lower cost, and stable in the oral environment	Bone cement and screw fixation, temporary crowns and bridges, obturators, retainers, and denture base material [42,43,44]
PLA	Polylactic acid	FDM	Biodegradable and environmentally friendly	Absorbable fracture internal fixation material, guided bone/tissue regeneration barrier membrane, and biological scaffold [51,54]
PLGA	Poly(lactic-co-glycolic acid)	FDM	Biodegradability and biocompatibility	Tissue engineering scaffolds, guided bone regeneration membrane, and drug-delivery carrier [58,59,60]
UV resin	Ultraviolet resin	SLA/DLP	High curing efficiency, low energy consumption, and low cost	Protheses in dental applications, retainers, dentures, and retainers [65,66]

**Table 3 polymers-15-02405-t003:** Metals for 3D printing in dentistry.

Abbreviation	Material	Applicable 3D Printing Technologies	Characteristics	Clinical Applications
Ti	Titanium and its alloys	SLS/SLM/EBM/DMLS	Extremely chemically stable oxide film, excellent mechanical properties, and outstanding biocompatibility, high cost, limited abrasion resistance, and potential toxicity [70]	Dental implants and scaffolds, dental crowns, and denture frameworks [69,78,79,80]
Co–Cr	Cobalt–chromium alloy	SLS/SLM/DMLS	Excellent mechanical properties, corrosion resistance, and good porcelain bonding properties, potentially causes allergic reactions [69,95,99,100]	RPD framework, 3-unit FPD framework, crowns, cast post and core [24,96,101,102,103,104]
SS	Stainless-steel	SLM	Superior physical properties, biocompatibility, excellent bactericidal ability, and low mechanical properties [107,108]	Implants and orthodontic components [109]
-	Magnesium alloy	SLM	Favorable mechanical properties and biocompatibility, difficult powder preparation [107,108]	Implants [111]

**Table 4 polymers-15-02405-t004:** Ceramics for 3D printing in dentistry.

Ceramic Material	Molecular Formula	Applicable 3D Printing Technologies	Characteristics	Clinical Applications
Glass	-	SLA/SLS	High mechanical strength, low electrical conductivity, high dielectric constant, good mechanical processing properties, chemical resistance, and thermal stability [123]	Hypersensitivity therapy, implant coatings, and bone regeneration in periodontal treatment [76]
Zirconia	ZrO_2_	SLA/SLS	Biocompatible, osteoconductive, high strength, reduced inflammatory response, high internal stress, easily suffers from cracks after sintering and high-volume shrinkage [135,137,138,129]	Preformed as root-canal piles, crowns and bridge restorations, and implant abutments [139]
Alumina	Al_2_O_3_	FDM/SLA/SLS	High mechanical strength, excellent chemical stability, good electrical insulation properties, and easy to fracture [145]	Implants, crowns, bridges, veneers, orthodontic brackets, dental composites, and bone cement materials

**Table 5 polymers-15-02405-t005:** Natural biopolymers for 4D printing in dentistry.

Natural Biopolymers	Representative SEM Figures	Applicable 4D Printing Technologies	Characteristics	Clinical Applications
Hyaluronic acid (HA)	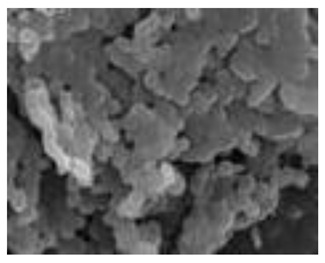 Reprinted with permission from Ref. [196]	Inkjet bioprinting/ Extrusion-based bioprinting (EBB)/ SLA-based bioprinting/ Laser-based bioprinting (LAB)	Biocompatibility, biodegradability, high mechanical properties, and high stability [169]	Printing products, promote tissue repair and wound healing, and effectively treat periodontitis and gingivitis [151,175]
Collagen	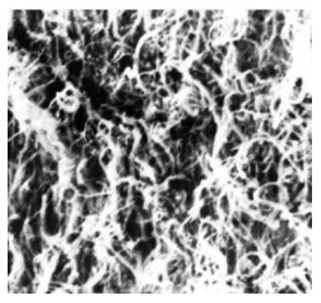 Reprinted with permission from Ref. [197]		Biocompatible and biodegradable	Cartilage tissue engineering [178]
Agarose	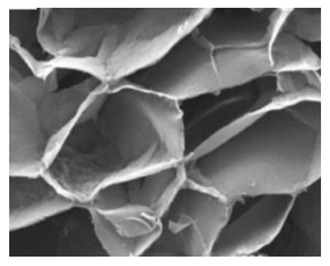 Reprinted with permission from Ref. [198]. Copyright 2019 Wiley.		Good gel properties, mechanical properties, biocompatibility, but inability to support cell growth [178,181,182,183]	Promote cell growth and enhance the mechanical properties of bone tissue [178,181,182,183]
Chitosan	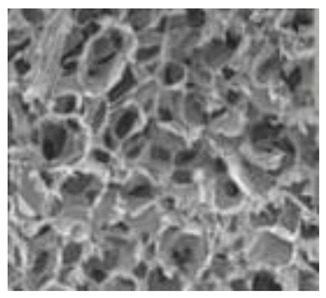 Reprinted with permission from Ref. [196]		Good biocompatibility, renewable ability, but low mechanical strength [185]	Widely used in prosthodontics, oral implantology, and endodontics [184]
Alginate	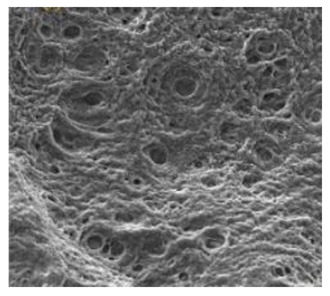 Reprinted with permission from Ref. [199]. Copyright 2018 Elsevier.		Good biocompatibility	Obtain conventional impressions and print 3D structures
Cellulose	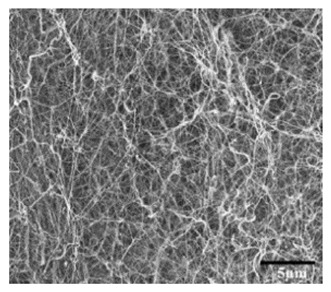 Reprinted with permission from Ref. [200]. Copyright 2006 Taylor & Francis.		Hydrophilic, as good insulating and anti-electrostatic properties [180,192,193]	Make tissue engineering scaffolds, wound dressings, and used in drug delivery [191]

**Table 6 polymers-15-02405-t006:** Cells applied in 4D printing in dentistry applications.

Cells	Full Name and Source	Representative SEM Figures	Targeted-Differentiated Tissue	Potential Applications in Dentistry
DPSCs	dental pulp stem cells	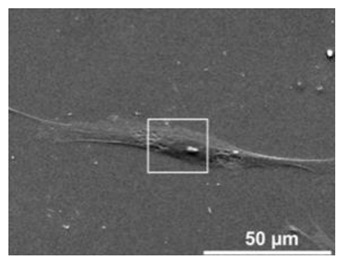 Reprinted with permission from Ref. [201]	osteoblasts, cementoblasts, odontoblats	alveolar bone regeneration, periodontal ligament regeneration, dental pulp regeneration [202]
SCAPs	stem cells from the apical papillae	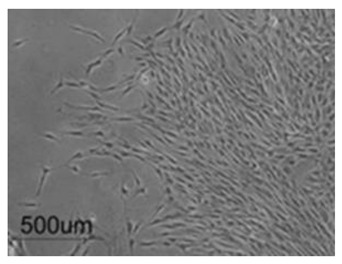 Reprinted with permission from Ref. [203]	odontoblasts, osteoblasts, neural cells, capillary network	endodontic treatment, pulp-dentin regeneration, bone regeneration, angiogenesis, neural regeneration and repair, periodontal tissue regeneration, bioroot engineering [204,205,206,207]
PDLSCs	periodontal ligament stem cells	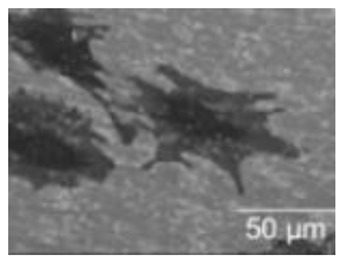 Reprinted with permission from Ref. [208]	neurons, fibroblasts, osteoblasts, cementoblasts	periodontal ligament regeneration, periodontal regeneration, mandibular defect repair, root regeneration [209]
DFCs	dental follicle cells	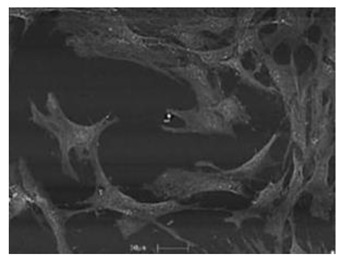 Reprinted with permission from Ref. [210]	osteoblasts, cementoblasts, and neurons	bone defects, pulp-dentin regeneration, and neural tissue regeneration [211]
GMSCs	gingival mesenchymal stem cells	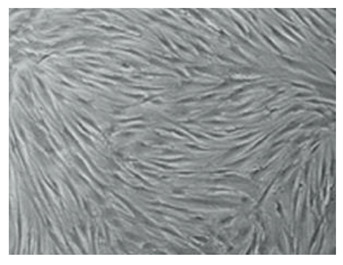 Reprinted with permission from Ref. [212]	osteoblasts, chondrocytes, neuro-like cells	periodontal ligament regeneration, facial nerve regeneration, osteogenesis, vascular regeneration [213,214]
SHEDs	stem cells from human exfoliated deciduous teeth	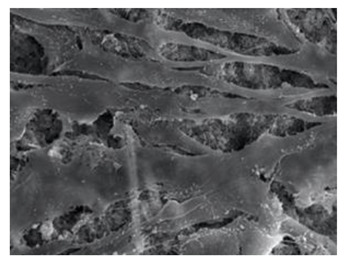 Reprinted with permission from Ref. [215]. Copyright 2017 Wiley.	osteoblasts, odontoblasts, neuro-like blasts,	pulp regeneration, bone regeneration [216]
hBMSC	human bone marrow stem cells	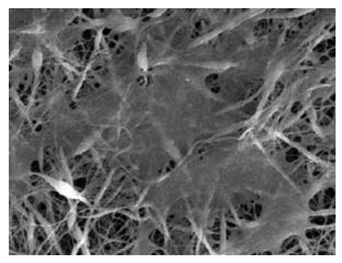 Reprinted with permission from Ref. [217]. Copyright 2003 IEEE.	osteoblasts	alveolar bone regeneration, bone regeneration, gene delivery [218]
HUVECs	human umbilical vein endothelial cells	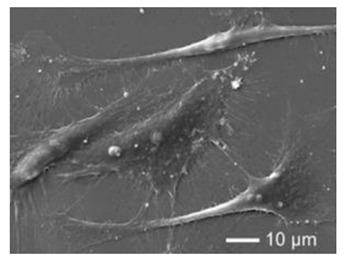 Reprinted with permission from Ref. [219]. Copyright 2007 Mary Ann Liebert, Inc.	pulp-dentin, microvessels, small blood vessels	cardio-vascularized tissue regeneration [220]
MC3T3-E1	osteoblast cell precursor MC3T3-E1	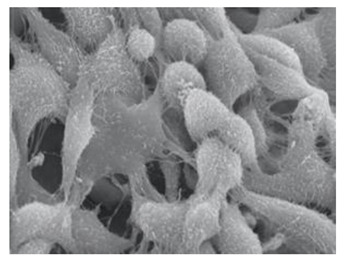 Reprinted with permission from Ref. [221]	osteoblasts, cementoblasts, bone tissue, nerve	alveolar bone regeneration [222]

**Table 7 polymers-15-02405-t007:** Growth factors for 4D printing in dentistry.

Abbreviation	Full Name	Representative SEM Figures	Characteristics	Clinical Applications
SDF-1	Stromal-derived factor 1	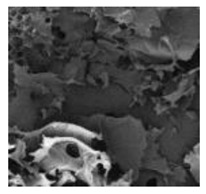 Reprinted with permission from Ref. [243]	Soluble signaling molecules Combine with printing materials to selectively release to control cell growth, value addition, differentiation, etc. [235,236]	Promote vascular regeneration and bone regeneration [237]
BMP-2	Bone morphogenetic proteins 2	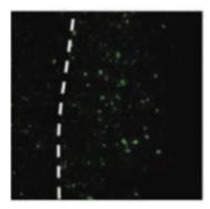 Reprinted with permission from Ref. [239]. Copyright 2023 Wiley.		Induce bone regeneration and bone healing
BMP-7	Bone morphogenetic proteins 7	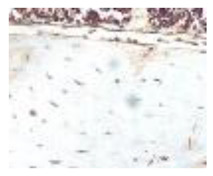 Reprinted with permission from Ref. [244]		
TGF-β1	Transforming growth factor-beta1	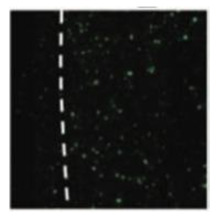 Reprinted with permission from Ref. [239]. Copyright 2023 Wiley.		Stimulate bioactivity and improve bone regeneration [239]
VEGF	Vascular endothelial growth factor	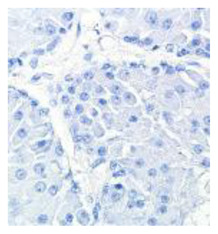 Reprinted with permission from Ref. [245]		Regulate cell migration, angiogenesis, and bone regeneration [238]

## Data Availability

The data presented in this study are available on request from the corresponding author.

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
