# Peer review of "Dental Materials Applied to 3D and 4D Printing Technologies: A Review"

_polymers, 2023, doi:10.3390/polym15102405_

Round 1

Reviewer 1 Report

In this Manuscript entitled “Dental Materials Applied to 3D- and 4D-Printing Technologies”, the authors classified, summarized, and illustrated dental materials for 3D/4D printing from a clinical perspective. The authors statistically analyzed the number of articles appearing on different databases like PubMed, Web of Science, Google Scholar, newspapers, magazines, books, and other sources. They classified materials into four major categories polymers, metals, ceramics, and biomaterials. Furthermore, the authors described 3D/4D printing materials, their properties, printing technologies, and scope of clinical applications. Future research directions have also been outlined by the authors.

Overall, some issues are associated with this review article, which need to be addressed before possible publication.

Please find the attached annotated file to see my comments.

Lastly, I would like to say “Polymers” journal publishes high-quality review articles related to 3D/4D printing. Based on my comments mentioned in the annotated file, the recommendation is Major Revision

The manuscript needs a thorough editing. Please see my comments in the annotated pdf file

Author Response

Thank you very much for your comments and suggestions. We have made revisions according to your comments and hope that they will be adequate for the acceptance of this manuscript.

Please see the attachment files.

Thank you,

Reviewer 2 Report

I think this review shall be published. However, I already see for instance, two review papers in similar themes:

Javaid, M., Haleem, A., Singh, R. P., Rab, S., Suman, R., & Kumar, L. (2022). Significance of 4D printing for dentistry: Materials, process, and potentials. Journal of Oral Biology and Craniofacial Research.

Khorsandi, D., Fahimipour, A., Abasian, P., Saber, S. S., Seyedi, M., Ghanavati, S., & Makvandi, P. (2021). 3D and 4D printing in dentistry and maxillofacial surgery: Printing techniques, materials, and applications. Acta biomaterialia122, 26-49.

Please state why your review in different from these two other. 

In many papers, they speak about the different materials you have mentioned, therefore, I think that information, instead of being given in the paper, I could be referenced, and use personal data and insight that you have from your lab for that. For example, picture of 3D printing dental parts with metals or ceramics. Showing also personal data in reviews is important.

Also, give insight and discussion about the present topic in another chapter of the paper. This must be the most important part. 

Good points: picture well presented and data organised. 

Author Response

(The authors gave the same response as above.)

Round 2

Reviewer 1 Report

It can be accepted now

Please see my previous comments

Reviewer 2 Report

It can be accepted in the present form.